# Transcriptional Regulation of Autophagy Genes via Stage-Specific Activation of CEBPB and PPARG during Adipogenesis: A Systematic Study Using Public Gene Expression and Transcription Factor Binding Datasets

**DOI:** 10.3390/cells8111321

**Published:** 2019-10-25

**Authors:** Mahmoud Ahmed, Trang Huyen Lai, Jin Seok Hwang, Sahib Zada, Trang Minh Pham, Deok Ryong Kim

**Affiliations:** Department of Biochemistry and Convergence Medical Sciences and Institute of Health Sciences, Gyeongsang National University School of Medicine, Jinju 527-27, Korea; ma7moud_sha3ban@hotmail.com (M.A.); tranghuyen20493@gmail.com (T.H.L.); cloud8104@naver.com (J.S.H.); s.zada.qau@gmail.com (S.Z.); phamminhtrang010895@gmail.com (T.M.P.)

**Keywords:** autophagy, transcription, adipocyte, differentiation, PPARG, CEBPB

## Abstract

Autophagy is the cell self-eating mechanism to maintain cell homeostasis by removing damaged intracellular proteins or organelles. It has also been implicated in the development and differentiation of various cell types including the adipocyte. Several links between adipogenic transcription factors and key autophagy genes has been suggested. In this study, we tried to model the gene expression and their transcriptional regulation during the adipocyte differentiation using high-throughput sequencing datasets of the 3T3-L1 cell model. We applied the gene expression and co-expression analysis to all and the subset of autophagy genes to study the binding, and occupancy patterns of adipogenic factors, co-factors and histone modifications on key autophagy genes. We also analyzed the gene expression of key autophagy genes under different transcription factor knockdown adipocyte cells. We found that a significant percent of the variance in the autophagy gene expression is explained by the differentiation stage of the cell. Adipogenic master regulators, such as CEBPB and PPARG target key autophagy genes directly. In addition, the same factor may also control autophagy gene expression indirectly through autophagy transcription factors such as FOXO1, TFEB or XBP1. Finally, the binding of adipogenic factors is associated with certain patterns of co-factors binding that might modulate the functions. Some of the findings were further confirmed under the knockdown of the adipogenic factors in the differentiating adipocytes. In conclusion, autophagy genes are regulated as part of the transcriptional programs through adipogenic factors either directly or indirectly through autophagy transcription factors during adipogenesis.

## 1. Introduction

Autophagy, a self-eating process, is involved in the development and the differentiation of various cell types including the adipocyte. Autophagy contributes to the adipocyte differentiation by reorganizing the intracellular components, forming the fat droplets and/or providing the energy source for the development of the phenotype [1]. Therefore, the knockdown of essential autophagy genes prevents the adipocyte differentiation and intervenes in the lipid accumulation within adipogenic cells [2]. The adipocyte differentiation is a well-understood process. A well defined transcriptional program is initiated and maintained by specific transcription factors. The 3T3-L1 mouse fibroblast is a key model for studying this process. When the pre-adipocyte is induced with a chemical cocktail containing MDI (1-Methyl-3-isobutylxanthine, Dexamethasone, and Insulin), the cell witnesses morphological and metabolic changes and is finally transformed into mature adipocyte [3].

Several studies reported functional links between the adipogenic factors and key autophagy genes. For example, the transcription factor CCAAT enhancer-binding protein beta (CEBPB) controls the expression of the autophagy-related 4b cysteine peptidase (*Atg4b*) [4]. A recent study suggested that peroxisome proliferator-activated receptor gamma (PPARG) regulates autophagy through NEDD4 E3 ubiquitin protein ligase (NEDD4) gene expression in thiazolidinediones-treated HepG2 cells [5]. Forkhead box o1 (FOXO1) regulates autophagy induction and it also negatively regulates Pparg, the master adipogenic factor [6,7]. Indeed, the inhibition of FOXO1 is thus necessary for the differentiation to proceed [7,8]. In a previous study from our laboratory, we reported the progressive activation of the *Foxo1* gene during the course of differentiation, to reach the highest expression in the mature adipocytes [9]. In addition, we identified a large number of differentially expressed autophagy genes between differentiated and undifferentiated adipocytes. The changes in gene expression were clustered into several groups that corresponded to the known stages of adipocyte differentiation. Therefore, the regulation of autophagy during the course of the differentiation seems to be more dynamic and stage-dependant. Only a few examples of direct and indirect regulation of autophagy genes have been studied.

We hypothesize that the autophagy process may be regulated by the same adipogenic factors either directly or indirectly through specific autophagy transcription factors. To investigate this hypothesis, we modeled the gene expression and transcription factors and co-factors binding at different stages of adipocyte maturation (Figure 1). Then, we studied the direct regulatory links between the adipogenic transcription factors and key autophagy genes as well as specific autophagy transcription factors. We were able to quantify the changes in the global expression of autophagy genes as the pre-adipocytes progressed toward the maturation stage. We considered changes at both the individual gene and the gene set levels. Then, we investigated the links between the master adipogenic transcription factors and the observed changes in gene expression. We tested whether the established links hold in the knockdown condition of the adipogenic factors. In addition, the role of transcription co-factors and histone modifications in modulating the function of the transcription factor was discussed.

## 2. Results

### 2.1. Autophagy Genes are Regulated as Part of the Transcriptional Program of the Adipocyte Differentiation

First, to model the changes in gene expression during the course of the adipocyte differentiation, we used datasets of RNA-Seq 3T3-L1 cells induced with the MDI cocktail and sampled at different time points (−48 to 240 h) (Figure 1). In agreement with the previous literature, the differentiation course was divided into non, early and late differentiation stages. This classification could explain the variance in expression (>50%) of all genes in multidimensional scaling (MDS) analysis (Figure 2 and Appendix A). Moreover, the variance in the expression of subsets of genes of interest was also explained by the stage of differentiation (Figure 2). Significant amounts of variance (85% and 87%) in the expression of autophagy genes and autophagy transcription factor genes were explained by the first two dimensions of the MDS (Appendix A). Together, the three-stage classification of differentiating cells explains significant amounts of the variance in gene expression. The subset of autophagy genes exhibits temporal changes that correspond to the differentiation course.

To characterize the changes in gene expression in the adipocytes in response to MDI induction, we applied the differential gene expression analysis to all genes after removing low-quality samples and low expressed genes. Significant changes in the global expression were observed when comparing early-differentiated samples to the non-differentiated and these changes were more pronounced in the late-differentiated samples (Appendix A). The changes are illustrated by the induction of adipogenesis (>1.5 log${}_{2}$ fold-change; FDR <0.2) and lipogenesis (>0.5 log${}_{2}$ fold-change; FDR < 0.2) markers such as *Pparg*, CCAAT enhancer-binding protein alpha/beta (*Cebpa*/*b*), ATP citrate lyase (*Acly*), fatty acid synthase (*Fasn*) and lipoprotein lipase (*Lpl*) (Appendix A). Moreover, several lipid metabolism-related gene sets such as lipid droplet and lipid storage were enriched in the early- and late-differentiating adipocytes compared to the pre-adipocytes (Appendix A).

The same pattern was observed in the expression of the subset of autophagy genes (Appendix A). Key autophagy gene markers such as beclin 1 (*Becn1*) was up-regulated (0.31 and 0.44 log${}_{2}$ fold-change; FDR <0.2) in both early- and late-differentiated cells. Other marker’s expression such as microtubule associated protein 1 light chain 3 beta (*Map1lc3b*) and sequestosome 1 (*Sqstm1*) was recovered (>0.5 log${}_{2}$ fold-change; FDR <0.2) in the late stages compared to pre-adipocytes (Appendix A). In addition to changes in the expression of individual gene markers, qualitative changes in subsets of the autophagy genes were observed. In particular, the subset of regulators of macroautophagy and autophagosome maturation were enriched between stages of differentiation (Appendix A).

### 2.2. Adipogenic Transcription Regulators Correlate with the Expression of Autophagy Genes

The genes encoding for the adipogenic transcription factors CEBPB and PPARG were one to two log${}_{2}$ folds up-regulated in the early stage of differentiation (Table 1). In the case of *Pparg*, the upward expression trend continued to the late stage. Known co-factors such as the mediator complex subunit 1 (*Med1*) and retinoid X receptor gamma (*Rxrg*) were also up-regulated in differentiating compared to the non-differentiated cells (Table 1). By dividing autophagy genes into up- and down-regulated groups, we were able to show the potential direction of the regulation if indeed some of these co-expression relations are functional. CEBPB was co-expressed (PCC >0.25) with down-regulated autophagy genes only in the early stage of differentiation (Figure 3). By contrast, *Pparg* was co-expressed (PCC >0.5) with the regulated genes in both directions and in both early and late stages (Figure 3). Moreover, the co-factors *Rxrg* and *Med1* showed a co-expression pattern with autophagy genes that is identical or opposite to that of *Pparg* in at least one of the differentiation stages (Figure 3).

### 2.3. Master Adipocyte Regulators CEBPB and PPARG Directly Target Key Autophagy Genes

Key autophagy genes such as *Map1lc3b* and *Sqstm1* were briefly down-regulated in non-induced and early differentiated adipocytes. The expression pattern was gradually reversed (0.45 and 0.95 log${}_{2}$ folds) over time and toward the late differentiation stage. *Becn1* was up-regulated (0.31 and 0.44 log${}_{2}$ fold-change; FDR <0.2) in both early- and late-differentiated cells (Figure 4a). The expression of the two adipogenic factors negatively correlates (PCC −0.2 to −0.5) with that of the three autophagy genes. Therefore, they may act as negative regulators or they would be de-recruited away from the regulatory regions of these genes at some point in the differentiation course. Only in the case of *Becn1*, the correlation is reversed with *Cebpb* and *Pparg* in the early and late-differentiation stages (Appendix A). Based on this information only, the two factors can work on *Becn1* cooperatively or consecutively at their most active stages.

To quantify the active binding of transcription factors on autophagy genes and its change over time, we performed differential peak binding to compare the binding in differentiating sample compared to the non-induced samples (Appendix A). CEBPB displays a similar active binding on *Becn1* and *Map1lc3b* (Figure 4b). At least one peak around the promoter region of the two genes is observed in non-differentiated cells (Figure 4d). These peaks were less enriched (<−1 log${}_{2}$ fold-change) in the early differentiation stage only to regain the enrichment to a similar expression in the late stage (Appendix A). PPARG show an opposing binding pattern on *Becn1* and *Sqstm1* (Figure 4b). The change in enrichment is strongest in *Sqstm1* where the binding peak in early differentiated cells decreases (−2.75 log${}_{2}$ fold-change) in the late stage (Figure 4c and Appendix A). The observed patterns suggest dynamic binding which might be a result of the availability or the distribution of the proteins over time. Unlike CEBPB, the binding of PPARG on the key autophagy genes may also occur in fibroblasts and macrophages (Appendix A).

### 2.4. CEBPB and PPARG Target Autophagy Transcription Factors Genes

With the exception of transformation-related protein 53 (*Trp53*), several autophagy transcription factor genes were found to be induced in one or both differentiation stages (Appendix A). *Foxo1*, X-box binding protein 1 (*Xbp1*) and transcription factor EB (*Tfeb*) were up-regulated (>1.5 log${}_{2}$ fold-change) overtime to reach the highest expression toward the end of the differentiation process (Figure 5a). Both adipogenic factors showed active binding patterns on at least one of the three autophagy transcription factor genes (Figure 4b). The strongest binding activity around the promoter areas of the targets was that of CEBPB on *Xbp1* and that of PPARG on *Tfeb* (Figure 4c,d and Appendix A). Both cases were accompanied by positive co-expression of the corresponding genes (Appendix A). The two factors seem to also have some binding activity around the promoter of *Foxo1* (Figure 4c,d). Most of these binding activities are probably tissue-specific since little to none was observed in other tissue types (Appendix A).

### 2.5. Auto-Regulation and Transcriptional Feedback Loops May Affect the Regulation of Autophagy

Although not directly related to the regulation of autophagy genes, the adipogenic transcription factors may target their own coding genes to form auto-regulation circuits or a feedback loops (Figure 6b). CEBPB targets the promoter region of its own gene (>1 log${}_{2}$ fold-change) and that of *Pparg* (<−0.7 log${}_{2}$ fold-change) (Figure 6d). The enrichment of the binding peak of CEBPB on its own promoter in the late stage of the differentiation is accompanied by a decrease in the gene expression suggesting a negative auto-regulation (Figure 6a). Together with the binding of PPARG on the promoter of *Cebpb* (0.56 log${}_{2}$ fold-change) they form a feedback loop (Figure 6c). The binding of PPARG on *Cebpb* and the reverse is similarly observed in macrophages (Appendix A).

### 2.6. Co-Factors and Histone Modifications Modulate the Functions of Adipogenic Transcription Factors

In addition to the changes in binding patterns and intensity of the adipogenic transcription factors, the occupancy of the adipogenic factors is associated with that of the co-factors and the histone modification markers. Specifically, PPARG associates with MED1 and RXRG in all stages of differentiation and all genomic binding regions (Appendix A). Similarly, PPARG correlates with H3k27ac and H3k4me1 in late differentiation stages and at different genomic locations (Appendix A). This is not exactly the case for CEBPB where the correlations with the co-factors and histone markers are less at the late-differentiation stage. The same pattern is confirmed by the correlation in occupancy between the transcription factors and other co-factors and histone modifications in autophagy genes. PPARG associates the most with MED1 and RXRG especially in late differentiated cells and at the promoter regions of this subset of genes (Figure 7 and Appendix A). On the other hand, CEBPB is closest to the co-factor E1A binding protein p300 (EP300) at the same stage and location (Figure 7 and Appendix A).

Functionally, MED1 seems to be the most active co-factor of the three. The change in the occupancy of the co-factor mimics that of PPARG on the latter’s autophagy targets in late-differentiating cells (Figure 8a) and CEBPB targets in early-differentiating cells (Figure 8b). In contrast, EP300 and RXRG occupancy changes only correlate with that of CEBPB or PPARG respectively (Figure 8a,b). The histone modification markers that had the strongest correlation with respect to the change in tags on autophagy genes were H3K27ac on CEBPB targets in the early differentiation stage and H3K4me1 on PPARG targets in the late-differentiation stage (Figure 8c,d).

### 2.7. Perturbing the Adipogenic Factors in Differentiating Adipocytes Disturbs the Expression of Autophagy Genes

We evaluated the role of CEBPB and PPARG in regulating the gene expression of key autophagy genes. We applied the gene differential expression analysis on two datasets of *Cebpb*- and *Pparg*-knockdown in adipocytes at different time point of the differentiation course (Appendix A). The knockdown of either factor resulted in significant (log${}_{2}$ fold-change <−75; FDR <0.2) changes in the expression of lipogenic genes such as *Acyl*, *Lpl* and/or *Fasn*. These changes were pronounced in the late-adipocytes (day 5) with the knockdown of *Pparg* (Table 2 and Figure 9). The effect of the knockdown of both genes was also reflected by the enrichment of several lipid metabolism, transport and storage gene sets (Appendix A). The disruption of the gene expression as a result of perturbing the adipogenic factor genes extended to other genes of interest.

For example, the expression of autophagy transcription factor gene *Foxo1* was up-regulated (0.84 log${}_{2}$ fold-change) by *Cebpb*-knockdown and down-regulated by *Pparg*-knockdown at day 2 (0.81 log${}_{2}$ fold-change) (Table 2 and Figure 9). Conversely, *Map1lc3* was down-regulated by *Cebpb*-knockdown (−0.54 log${}_{2}$ fold-change) and up-regulated (0.93 log${}_{2}$ fold-change) by *Pparg*-knockdown (Table 2 and Figure 9). In addition, several autophagy gene sets including those representing autophagosome assembly and maturation were enriched by the knockdown of the adipogenic factor genes (Appendix A). Finally, the knockdown of *Cebpb* significantly reduced (<−0.65 log${}_{2}$ fold-change) the expression of the other adipogenic transcription factor gene *Pparg* in the non-induced adipocyte and four hours after induction alike (Table 2 and Figure 9a). Removing *Pparg* on the other hand had only slight effect on the other adipogenic factor genes *Cebpa* and *Cebpb* and only on day 2 of the differentiation (Table 2 and Figure 9b).

## 3. Discussion

We found that, key autophagy markers such as *Map1lc3b*, *Becn1* and *Sqstm1* were regulated starting from the second day after induction of the adipocyte differentiation and reached the highest expression in the mature adipocytes. Some of these key genes were targeted directly by CEBPB and/or PPARG. Other key autophagy genes maybe regulated by the adipogenic transcription factors indirectly through other transcription factors such FOXO1, XBP1 and TFEB. Finally, the binding of the adipogenic factors on these genes is associated with the binding of other co-factors and histone modifications. Figure 10 depicts a summary diagram of the findings in this study. These findings were investigated under the knockdown of the adipogenic factor genes in differentiating adipocytes at the relevant time points.

Autophagy is important in the early stage of differentiation for lipid accumulation [2,10,11]. The knock-down of *Atg7* impaired the differentiation process [2]. Our analysis suggests a progressive activation of autophagy genes starting at day 2 after induction as indicated by *Map1lc3b*, although still lower than its expression in the pre-adipocytes, which continues into the maturation stage (Appendix A). Other studies suggested the activation of *Atg4b* through CEBPB [4] and the inhibition of SQSTM1 which enhances the ERK activation at day 2 [12,13]. Both events were observed in our analysis (Table 1). However, we did not observe a strong binding of CEBPB on any of the autophagy-related genes. Therefore, we investigated the binding of adipogenic factors on other key autophagy regulators.

Several studies suggested the direct regulation of autophagy by adipogenic transcription factors. The expression of *Map1lc3b* and *Sqstm1* was directly controlled by CEBPB in hepatocytes [14]. CEBPG along with three other factors regulated the expression of CREB regulated transcription coactivator 2 (*TORC2*) in bovine adipocytes, which promotes autophagy [15,16]. Similarly, PPARG regulates autophagy through NEDD4 in HepG2 cells [5]. In agreement with the previous literature, we found that CEBPB binds around the promoter areas of *Map1lc3b* and *Sqstm1* genes in addition to *Becn1* although the binding intensity is more apparent after day 2 (Appendix A and Figure 4b,d). The latter also has binding sites for PPARG on which the occupancy of the transcription factor was the reverse of that of CEBPB over time (Figure 4b,c).

Other autophagy genes including autophagy-related genes that are under control of transcription factors TFEB, FOXO1 and/or XBP1 may be only indirectly regulated by adipogenic factors [17,18,19]. Both CEBPB and PPARG has binding sites on the genes encoding these three factors and appears to be actively regulated over time (Appendix A and Figure 5). Functionally, PPARG binding effect may be opposed to that of CEBPB on these genes as they have opposite binding patterns accompanied by up-regulation of their coding genes.

FOXO1 regulates adipocyte differentiation [7]. It inhibits the expression of *Pparg* therefore it should be deactivated early in the differentiation process to allow for the adipogenic factor activation [8]. Our analysis suggests that *Foxo1* is turned on at some point in the differentiation course and continues to be expressed in the mature adipocytes (Appendix A and Figure 5a). This might be explained by the combined binding of the two adipogenic factors CEBPB and PPARG on the gene and their change of occupancy over the course of differentiation (Figure 5). Here, the expression of *PPARG* plateaus and that of *Foxo1* continues to rise.

CEBPB was reported to induce *Pparg* indirectly through CEBPA [20]. However, there seems to be an alternative mechanism to this induction as it happens even in the absence of the CEBP proteins [21,22]. Our analysis suggests multiple binding sites of CEBPB around the promoter of *Pparg*, this might explain the induction at lease in the case of absent CEBPA (Figure 6b,d). Together with the observed binding of PPARG on CEBPB forms a transcriptional loop that might enforce this induction mechanism (Figure 6b,c). Finally, CEBPB appears to bind to its own promoter to auto-regulate its own expression (Appendix A and Figure 6b,d). The binding of the two factors on the *Cebpb* promoter may explain the timely reduction in the expression of the gene as they reach their highest occupancy toward the end of the differentiation course.

A study reported a positive correlation between PPARG and the co-factor MED1 but not CBP in response to rosiglitazone in differentiating adipocytes [23]. Here, we investigated the possibility of this recruitment happening during the natural course of differentiation, whether it is shared by other transcription factors and co-factors. We found that the PPARG occupancy on autophagy genes correlates with the occupancy of MED1, RXRG, and EP300 (Appendix A). CEBPB occupancy only correlated with these factors in the pre-adipocytes (Appendix A). In both cases, the locations of the peaks had no effect on the correlation (Appendix A). The occupancy of the co-factors MED1 and RXRG correlated the most with PPARG while EP300 correlated with CEBPB across stages and genomic regions (Figure 7 and Appendix A). However, the change in occupancy of PPARG and CEBPB on their own targets was positively correlated with the change of occupancy of MED1 and RXRG or MED1 and EP300 in early or late-differentiation respectively (Figure 8a,b).

CEBPB and PPARG change in occupancy on their respective autophagy targets were mimicked by H3K27ac and H3K4me1 (Figure 8c,d). Both histone modifications were reported to mark active areas of the chromatin; the active genes in case of H3K27ac and active enhancers in the case of H3K4me1 [24,25]. This is consistent with the reported non-transcriptional role of PPARG which helps to reorganize the chromatin in the differentiating cells [26]. In other words, the areas of the chromatin which contain genes coding for autophagy proteins are activated as evidenced by the changes in the active chromatin markers (Appendix A and Figure 8b,c). These changes are associated with parallel changes in the adipogenic factor occupancy much like their non-transcriptional role in the spatial reorganization of adipogenic genes.

Although the knockdown of *Cebpb* or *Pparg* in the manner analyzed in this study may not reveal the functional effect on the phenotype, it provides confirmation of the mechanistic role of the transcription factors in regulating the expression of their own genes as well as of autophagy genes. Removing either transcription factors significantly reduced the expression of important lipogenic genes such as *Acyl* and *Lpl* (Table 2 and Figure 9). Similarly, the knockdown of *Cebpb* significantly reduced the expression of *Pparg* in non-induced and induced adipocytes (Table 2 and Figure 9a). As expected, the opposing regulation patterns of the adipogenic factors on key autophagy genes such as *Foxo1* and *Map1lc3b* were reflected in their knockdown patterns (Figure 9). However, we also expect some gene expression changes to be an indirect consequence of removing or relocating the factors to other regions. In addition, some important genes such as *Becn1* and *Tfeb* didn’t have probes corresponding to them in the microarrays dataset and we were not able to confirm their regulation under the *Pparg*-knockdown condition.

lThe 3T3-L1 cell line is a useful adipocyte model. It has many applications in obesity and insulin resistance research such as lipid synthesis, white vs. brown adipose tissue development, insulin-sensitizing drug action [27,28,29]. However, the findings based on this model should be considered to the extent that they reflect the development and the biology of the mature adipocytes. Therefore, the validation of such findings in primary human adipocytes or fat tissue isolates is important and is a natural extension to the current study. In addition, detailed experimentation may be required to study the conditions under which the regulatory links reported here result in qualitative changes in the gene expression and protein level of the targets. Finally, future studies are needed to confirm that the changes at the transcription level translate into autophagy activity.

Studying the reported findings under perturbation of the differentiation course would provide useful information. For example, it might be interesting to study the course of differentiation under autophagy inhibition or absence to quantify the changes in its gene expression profiles. Moreover, the conditional knockdown of CEBPB and/or PPARG in early and late-differentiation stages would provide details about the functional roles of the regulatory links reported in this study. Finally, the contributions of the direct and indirect regulation of the adipogenic transcription factors to autophagy genes can be determined by conditionally modifying or removing the intermediate autophagy transcription factors. Indeed, evaluating these findings under chemical and genetic modification of the differentiation course is a natural extension of this study and a goal for future investigation.

Using publicly available datasets from different sources represents a challenge to the current investigation. We addressed this challenge by manually curating the datasets, applying proper quality control measures and using proper quantitative analysis. Moreover, depending on the availability of the data, some of the phenotypes of interest might have a small number of samples, low time resolution or be missing altogether. In this study, we used quantitative models and reported all findings with an attached degree of uncertainty to reflect the degree of confidence in the data they were drawn from. Finally, we abstracted the time course of the differentiation into meaningful stages and key time points to address the missing data issues.

## 4. Materials and Methods

### 4.1. Pre-Adipocyte 3T3-L1 Differentiation Protocol

The data included in this study used the MDI (1-Methyl-3-isobutylxanthine, Dexamethasone, and Insulin) differentiation protocol with minimal variations [3]. The induction starts two days post-confluence and lasts for one or two weeks with changing the media at fixed intervals. At the end of it, the efficiency of the induction can be evaluated using adipogenic markers or lipid staining. The differentiation course was divided by time into three stages (non, 0 h and before; early, after 0 to 48 h; and late, after 48 to 260 h). The grouping criteria were previously suggested by others and devised from the data itself as it explains the largest amount of variance.

### 4.2. Data Collection, Pre-Processing, and Processing

The data collection strategy, pre-processing and processing of the datasets used in this study is documented in two Bioconductor packages curatedAdipoRNA and curatedAdipoChIP (submitted). Briefly, we surveyed the gene expression omnibus (GEO) and the sequence read archive (SRA) repositories for high-throughput sequencing data of MDI induced 3T3-L1 pre-adipocyte at different time points. The data were obtained from GEO/SRA in the form of raw reads. In total, 98 RNA-Seq and 207 transcription factor, co-factor and histone modification markers ChIP-Seq samples were included.

For RNA-Seq (n = 66, subset), the raw reads were aligned to mm10 mouse genome using HISAT2 [30]. featureCounts was used to count the reads aligned to the known genes [31]. For ChIP-Seq (n = 96, subset), the raw reads were aligned to mm10 using BOWTIE2, peaks were called and signal tracks were built using MACS2 and reads counted in a peak set of replicated peaks across samples were obtained using BEDTOOLS [32,33,34]. The peak set was annotated and peaks were assigned to the nearest gene using ChIPseeker [35]. FASTQC was used to assess the quality of the raw reads [36]. The phenotype data of the samples were manually curated to match the time point, stage of differentiation, the binding factors/markers and the ChIP antibodies (Appendix A).

Another dataset of CEBPB and PPARG ChIP-Seq samples in different tissue was used to test the binding specificity of the adipogenic transcription factors. The Cistrome Data Browser was searched for the adipocyte, fibroblast of macrophage (*Biological Sources*) and CEBPB/PPARG (*Factors*). The results were manually reviewed to select the ChIP-Seq samples (n = 11) of the two factors in these tissues (Appendix A). The data were obtained in the form of processed signal tracks. Two datasets of 8 RNA-Seq and 18 microarrays samples of *Cebpb*- and *Pparg*-knockdown in differentiating adipocytes was obtained, processed and analyzed (Appendix A). 3T3-L1 cells were transected with either shRNA against *Cebpb* or siRNA against *Pparg*. Transected cells were subsequently differentiated into mature adipocytes using the previously described protocol. Total RNA was harvested and the gene expression was profiled at zero/four hours or zero/two/five days post-induction using RNA-Seq or microarrays for *Cebpb*- and *Pparg*-knockdown cell, respectively.

### 4.3. Gene Expression Analysis

The gene counts were used to apply the differential expression analysis using DESeq2 [37]. The samples were divided into three groups corresponding to the stages none, early and late-differentiation. Three contrast groups were applied (early vs. non, late vs. non and late vs. early). For each gene, a log${}_{2}$ fold-change and a false discovery rate (FDR) were calculated. Genes with absolute log${}_{2}$ fold-change >1 and FDR <0.2 were considered significantly expressed and depending on the sign of the fold-change to be up- or down-regulated in one or more of the three contrasts. Using the gene counts a multidimensional scaling (MDS) analysis was applied to all, autophagy and adipogenic transcription factor genes using cmdscale (base R) [38]. The differential gene expression was applied to the *Cebpb*-knockdown RNA-Seq and *Pparg*-knockdown microarrays datasets by comparing the expression in knockdown (KD) vs. control at each time point using DESeq2 and limma [37,39].

The biological process gene ontology (GO) term autophagy (GO:0006914) was used to define the genes with known functions in autophagy (n = 158) [40]. The molecular function term DNA-binding transcription factor activity (GO:0003700) was used to define genes with known functions as transcription factors (n = 745). The intersection of the two terms were the six autophagy transcription factors genes. The GO annotations were accessed using org.Mm.eg.db [41]. Other genomic annotation such as gene coordinates and identifiers were accessed using TxDb.Mmusculus.UCSC.mm10.knownGene [42].

### 4.4. Gene Set Enrichment Analysis

Several autophagy and lipid metabolism GO terms and their gene products were identified. The lists of differentially expressed genes were used to determine the over-represented gene sets in each comparison. For each gene set term, the ratio of the differentially expressed genes to the number of the genes in the term was compared to the total number of the differentially expressed genes to the total number of genes. A two-by-two table was constructed from the list and tested to calculate a *p*-value. The packages goseq and clusterProfiler were used to apply this analysis on the RNA-Seq and the microarrays datasets, respectively [43,44].

### 4.5. Gene Co-Expression Analysis

The gene counts were transformed using variance stabilization transformation (VST). The transformed counts were used to calculate the Pearson’s correlation coefficient (PCC) as a measure of co-expression between each pair of genes in each condition. The coefficients were transformed into z-scores. The difference in z-score and the variance between every two conditions is used to calculate *p*-value. The z-scores from the original and permuted datasets are used to calculate empirical *p*- and q-values for multiple testing adjustment. Pairs of genes were considered differentially co-expressed when FDR <0.2. DGCA was used to apply this analysis [45].

### 4.6. Peak Binding Analysis

After removing the low-quality ChIP-Seq samples and the low count peaks in the peak set, the reads count in peaks were used to apply the differential peak binding analysis using DESeq2 [37]. The samples were divided into three groups based on the time point (non, 0 h and before; early, after 0 to 48 h; and late, after 48 to 260 h). Three contrast groups were applied (early vs. non, late vs. non and late vs. early). For each peak, a log${}_{2}$ fold-change and an FDR were calculated. Peaks with absolute log${}_{2}$ fold-change >0.5 and FDR <0.2 were considered significantly expressed and depending on the sign of the fold-change to be up- or down-regulated in one or more of the three contrasts.

### 4.7. Occupancy and Affinity Analyses

The reads count in peaks were aggregated by gene to find the total occupancy of the factors and histone modification markers. The occupancy of the factors and histone markers were grouped by genomic annotations (Promoter, 3’ Untranslated Region (UTR), 5’ UTR or other). The occupancy correlation of each sample was calculated using the PCC of all pairs reads count in peaks, total occupancy or occupancy grouped by genomic annotation. Promoters were defined as ±3 kb around the transcription start site (TSS) of each transcript. The signal tracks were used to visualize the peaks in the promoter regions of selected genes as an enrichment score relative to known controls.

### 4.8. Data Management, Transformation and Visualization

The processed data were stored mainly in the ExpressionSet, SummarizedExperiment and bigwig formats [46,47]. GEOquery was used to obtain the microarrays data [48]. collapseRows (WGCNA), tidyverse, GenomicRanges and rtracklayer were used for data transformation [49,50,51,52]. The ggplot2, GGally, xtable, ComplexHeatmap and EnrichedHeatmap were used for data visualization [53,54,55,56,57].

### 4.9. Source Code and Reproducibility

The input, output and the tools used in each step of the study workflow is depicted in Figure 1. The software environment where the full analysis and manuscript production were done is available as a docker container for reproducibility (https://hub.docker.com/r/bcmslab/autoreg). The analysis was conducted in R (3.5) using Bioconductor (3.6) and other R packages referred to in the relevant subsections [46,58]. The source code for conducting this analysis is maintained on (https://github.com/BCMSLab/autoreg). The code for reproducing the figures and tables presented in this manuscript is available at (https://github.com/BCMSLab/auto_adipo_diff).

## 5. Conclusions

Autophagy genes are regulated as part of the differentiation course of the adipocytes. This regulation is driven by adipogenic transcription factors such as CEBPB and PPARG. The adipogenic factors target key autophagy genes such as *Becn1*, *Map1lc3b* and *Sqstm1*. Moreover, other autophagy genes are regulated indirectly through autophagy transcription factors *Foxo1*, *Tfeb* and *Xbp1*. Co-factors such as MED1 actively contribute to the transcription factors binding on autophagy genes. Other co-factors such as RXRG and EP300 associate specifically with PPARG or CEBPB respectively. H3K4me1 and H3K27ac markers also associate with the adipogenic factors binding sites indicating non-transcriptional roles such as organizing the chromatin regions containing autophagy genes. 

## Figures and Tables

**Figure 1 cells-08-01321-f001:**
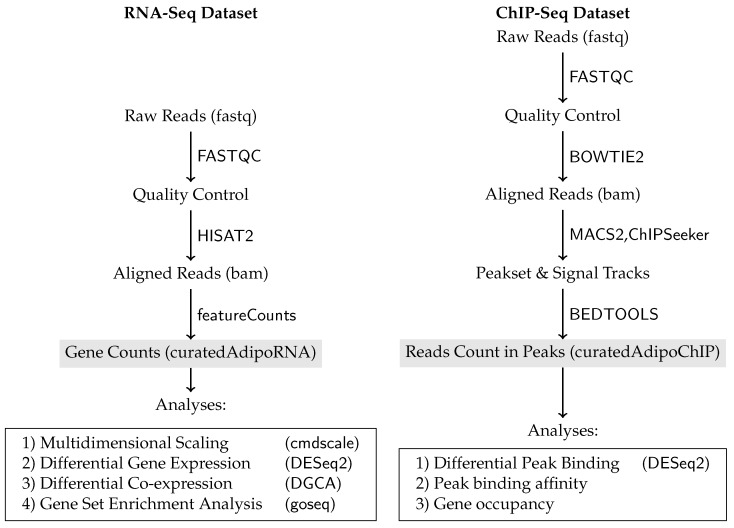
Data pre-processing, processing and analyses’ workflow of RNA-Seq and ChIP-Seq datasets.Pre-processing, processing and analyses’ workflow

**Figure 2 cells-08-01321-f002:**
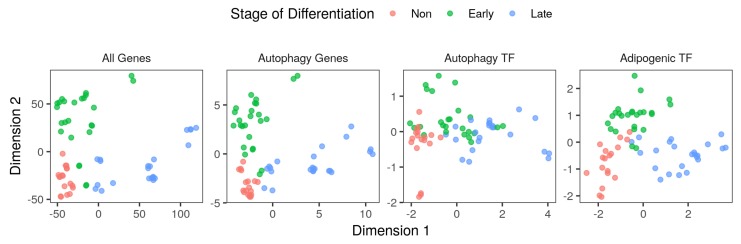
Multidimensional scaling analysis of gene expression. The transformed read counts of 15,786 genes coding, 158 autophagy genes, six genes encoding autophagy transcription factors (*Foxo1*, *Irf8*, *Tfeb*, *Trp53*, *Xbp1* and *Zkscan3*) and five genes encoding adipogenic transcription factors (*Cebpb*, *Ep300*, *Med1*, *Pparg* and *Rxrg*) were used to perform multidimensional scaling (MDS). Counts were extracted from RNA-Seq samples (n = 66) of MDI-induced 3T3-L1 cells in three differentiation stages; 18 non (red), 25 early (green) and 23 late (blue) differentiating samples.

**Figure 3 cells-08-01321-f003:**
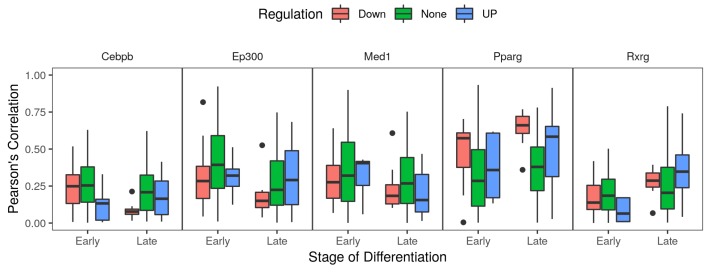
Co-expression of autophagy products with adipogenic transcription factors. The read counts of five genes coding for adipogenic transcription factors/co-factors and 158 autophagy genes were used to calculate the co-expression values of each of the factors with autophagy genes. Pearson’s correlation co-efficient were calculated from RNA-Seq samples (n = 66) of MDI-induced 3T3-L1 cells in three differentiation stages; non, early and late differentiating cell. Genes were stratified by their differential expression into down- (red), none- (green) or up-regulated (blue).

**Figure 4 cells-08-01321-f004:**
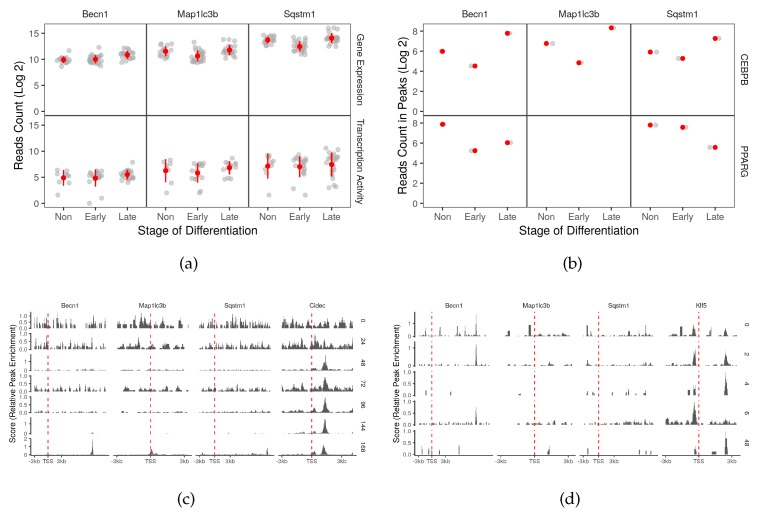
Key autophagy gene targets of the adipogenic transcription factors. (**a**) The reads count of three key autophagy genes in RNA-Seq samples (n = 66) were used to represent the gene expression. The reads count in peaks of POLR2A in genomic region of key autophagy genes from ChIP-Seq samples (n = 12) was used to represent the transcription activity. (**b**) The reads count in peaks of CEBPB and PPARG ChIP-Seq samples (n = 9 and 13) was used to represent peak enrichment. The Peaks from the (**c**) PPARG and (**d**) CEBPB ChIP-Seq samples were visualized by the time point and position around the transcription start site (±3 kb). *Cidec* and *klf5* were used as positive control genes.

**Figure 5 cells-08-01321-f005:**
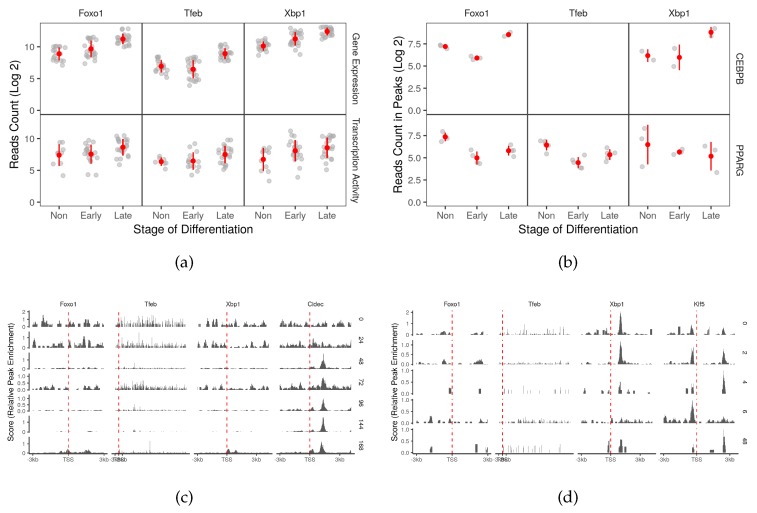
Autophagy transcription factor gene targets of the adipogenic transcription factors. (**a**) The reads count of three autophagy transcription factor coding genes in RNA-Seq samples (n = 66) were used to represent the gene expression. The reads count in peaks of POLR2A in genomic region of three autophagy transcription factor coding genes from ChIP-Seq samples (n = 12) was used to represent the transcription activity. (**b**) The reads count in peaks of CEBPB and PPARG ChIP-Seq samples was used to represent peak enrichment. The Peaks from the (**c**) PPARG and (**d**) CEBPB ChIP-Seq samples were visualized by the time point and position around the transcription start site (±3 kb). *Cidec* and *klf5* were used as positive control genes.

**Figure 6 cells-08-01321-f006:**
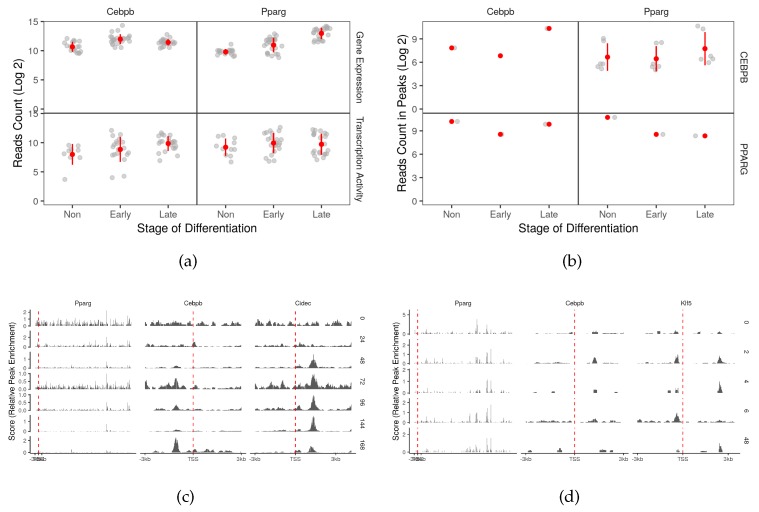
Auto-regulation and feedback loops of the adipogenic transcription factors. (**a**) The reads count of two adipogenic transcription factor coding genes in RNA-Seq samples (n = 66) were used to represent the gene expression. The reads count in peaks of POLR2A in genomic region of two adipogenic transcription factor coding genes from ChIP-Seq samples (n = 12) was used to represent the transcription activity. (**b**) The reads count in peaks of CEBPB and PPARG ChIP-Seq samples was used to represent peak enrichment. The peaks from the (**c**) PPARG and (**d**) CEBPB ChIP-Seq samples were visualized by the time point and position around the transcription start site (±3 kb). *Cidec* and *klf5* were used as positive control genes.

**Figure 7 cells-08-01321-f007:**
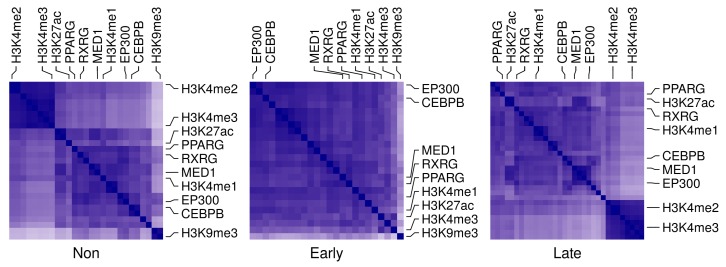
Correlation in occupancy between adipogenic transcription factors and histone markers on autophagy genes at different stages. The total number of reads in peaks of each ChIP-Seq sample (n = 84) was used to represent the factor/histone marker occupancy. Pearson’s correlation coefficient (PCC) was calculated for each pair of samples. The PCC value is between 0 (white) and 1 (blue). Samples were categorized by differentiation stage; non, early or late in that order.

**Figure 8 cells-08-01321-f008:**
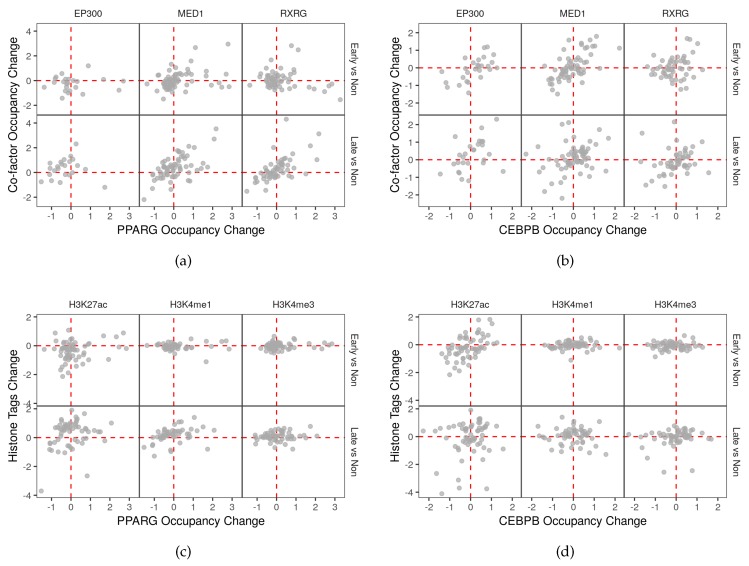
Change in occupancy of transcription factors and co-factors and histone marker tags. The total count of reads in peaks of autophagy genes (n = 158) of ChIP-Seq samples (n = 75) of transcription factors, co-factors and histone modification was used calculate the occupancy. The change in occupancy of (**a**,**c**) PPARG and (**b**,**d**) CEBPB in early or late differentiation vs. the non-differentiated samples were plotted against the change in occupancy of co-factors (**a**,**b**) or histone modification tags (**c**,**d**). Points represent the fold-change in occupancy between stages of differentiation.

**Figure 9 cells-08-01321-f009:**
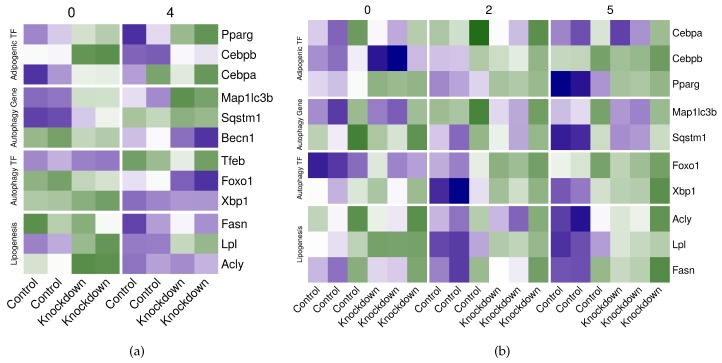
Expression of key adipogenic, autophagy and lipogenic genes in *Cebpb* or *Pparg*-knockdown adipocytes. (**a**) Reads count of selected genes from RNA-Seq samples (n = 8) of *Cebpb*-knockdown differentiating adipocytes at 0 and 4 h were scaled and shown as color values (low, green; high, blue). (**b**) Probe intensity of selected genes from microarray samples (n = 18) of *Pparg*-knockdown differentiating adipocytes at 0, 2 and 5 days were scaled and shown as color values (low, green; high, blue). Genes are indicated by symbols in rows and grouped in four categories (Adipogenic TF, Autophagy Gene, Autophagy TF or Lipogenesis). Sample group (knockdown vs. control) is indicated in rows and grouped by time points (0 or 4 h; and 0, 2 or 5 days).

**Figure 10 cells-08-01321-f010:**
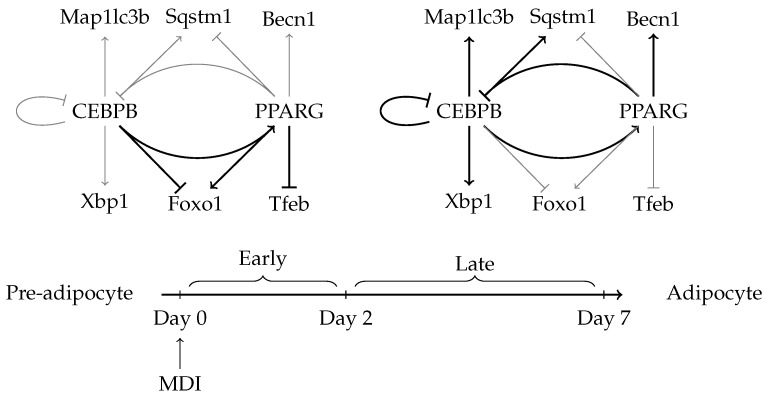
Direct and indirect regulation of autophagy genes by adipogenic transcription factors. A constructed model of the transcriptional activity of two adipogenic factors (CEBPB and PPARG) on their corresponding genes, key autophagy genes (*Map1lc3*, *Becn1* and *Sqstm1*) and autophagy transcription factors (*Xbp1*, *Foxo1* and *Tfeb*). The direction of regulation (←, induction; or ⊢, repression) is inferred based on the availability of the factor in each stage and the co-expression with the target gene. The change in occupancy and the binding intensity is indicated by the edge type (─, high; or ─, low).

**Table 1 cells-08-01321-t001:** Significant differentially expressed genes of adipogenic and autophagy transcription factors. FC, fold-change; SE, standard error.

Category	Gene	Early vs. Non		Late vs. Non		Late vs. Early	
		FC	SE	FC	SE	FC	SE
Adipogenic TF	Cebpb	1.5	0.19			−1.3	0.18
	Med1	0.3	0.1	0.28	0.1		
	Pparg	1.55	0.25	2.76	0.25	1.21	0.23
	Rxrg			7.89	0.72	6.89	0.63
Autophagy TF	Foxo1	1.1	0.22	2.03	0.22	0.92	0.2
	Tfeb			1.66	0.26	1.65	0.24
	Trp53	−0.52	0.18	−1.25	0.19	−0.73	0.17
	Xbp1	1.44	0.17	1.87	0.17	0.44	0.16
	Zkscan3			0.59	0.12	0.41	0.11
Autophagy Gene	Atg4b	0.38	0.07			−0.46	0.06
	Becn1	0.31	0.08	0.44	0.09	0.13	0.08
	Map1lc3a	−0.5	0.14			0.62	0.13
	Map1lc3b	−0.72	0.17	−0.27	0.17	0.45	0.15
	Sqstm1	−1.02	0.15			0.91	0.14
	Ulk1			0.74	0.15	0.55	0.14

**Table 2 cells-08-01321-t002:** Significant differentially expressed genes in *Cebpb* or *Pparg* knockdown adipocytes. KD, knockdown; h, hour; d, day; FC, fold-change; SE, standard error.

Category	Gene	Cebpb KD vs. Control				Pparg KD vs. Control					
		0 h		4 h		0 d		2 d		5 d	
		FC	SE	FC	SE	FC	SE	FC	SE	FC	SE
Adipogenic TF	Cebpb	−2.61	0.17	−1.94	0.17			−0.28	0.11		
	Pparg	−0.65	0.17	−1.19	0.27	−2.27	0.18	−1.92	0.19	−2.53	0.1
	Cebpa							0.29	0.11		
Autophagy Gene	Map1lc3b	−0.54	0.12	−0.67	0.19			0.93	0.3		
	Sqstm1	−0.58	0.11								
Autophagy TF	Foxo1	0.41	0.19	0.84	0.2			−0.81	0.26		
	Xbp1	−0.3	0.15					−0.88	0.09	−0.62	0.08
Lipogenesis	Acly	−0.75	0.11							−0.74	0.14
	Lpl	−2.54	0.37	−2.25	0.19	−2.18	0.21	−1.72	0.08	−1.36	0.1
	Fasn									−0.91	0.31

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
