# Peer review of "Transcriptional Regulation of Autophagy Genes via Stage-Specific Activation of CEBPB and PPARG during Adipogenesis: A Systematic Study Using Public Gene Expression and Transcription Factor Binding Datasets"

_cells, 2019, doi:10.3390/cells8111321_

Round 1
Reviewer 1 Report
This study showed that autophagy genes are regulated as part of the transcriptional programs through adipogenic factors either directly or indirectly through autophagy transcription factors during adipogenesis.
Evidence has shown that adipose tissue is an active endocrine organ, capable of secreting many cytokines, often referred to as adipokines, that can promote inflammation and interfere with insulin action [1]. Furthermore, some studies have shown that subcutaneous and visceral fat are biologically distinct, with visceral fat demonstrating far greater pro-inflammatory characteristics than subcutaneous fat [1]. Recently, it’s identified that C/EBP, XBP1, INSM1 and ZNF263 regulate TORC2 gene as activators in the promoter region [2]. Therefore, TORC2 gene, a member of the transducer of the regulated cyclic adenosine monophosphate (cAMP) response element binding protein gene family, is potentially a positive regulator of adipogenesis. [2]. Another, it’s suggested that PPARγ plays a novel role in linking glucose metabolism and protein homeostasis through NEDD4-mediated effects on the autophagy machinery [3].
Authors are kindly requested to emphasize the current concepts about these issues in the context of recent knowledge and the available literature. This articles should be quoted in the References list. In addition, figures and tables should be made readable.
References
Should visceral fat be reduced to increase longevity? Ageing Res Rev. 2013 Sep; 12 (4): 996-1004. doi:10.1016/j.arr.2013.05.007. Function and Transcriptional Regulation of Bovine TORC2 Gene in Adipocytes: Roles of C/EBP, XBP1, INSM1 and ZNF263. Int J Mol Sci. 2019 Sep 4; 20 (18). pii: E4338. doi: 10.3390/ijms20184338. PPARγ induces NEDD4 gene expression to promote autophagy and insulin action. FEBS J. 2019 Aug 18. doi:10.1111/febs.15042.Author Response
We fixed some typos and grammars throughout the manuscript. Also, we added a reference to report PPARG regulation of autophagy through NEDD4 and discussed it. However, we don’t see how the other two suggested references are very relevant to our study.
Reviewer 2 Report
The study by Ahmed et al. thoroughly and convincingly establishes the relationship between autophagy-related transcriptional regulation on the one hand, and adipocyte differentiation on the other. The study design, methods and results are clearly presented. My only concern is that the entire study seems to have been based on a single cell line. It would be great if at least some of the major findings could be validated in primary human pre-adipocytes before publication.
Author Response
1) The current study is entirely bound to the biology of this particular cell line and could only generalize to other kind of adipocytes to the extent of similarity in the underlying biology. However, extending the study to how ask similar or different the 3T3-L1 to human pre-adipocytes is beyond our goal.
2) The availability of data on human pre-adipocytes time course experiments is also a limiting factor.